# Effects of Combined Training Programs in Individuals with Fibromyalgia: A Systematic Review

**DOI:** 10.3390/healthcare11121708

**Published:** 2023-06-11

**Authors:** Mónica Sousa, Rafael Oliveira, João Paulo Brito, Alexandre Duarte Martins, João Moutão, Susana Alves

**Affiliations:** 1Sports Science School of Rio Maior, Polytechnic Institute of Santarém, 2040-413 Rio Maior, Portugal; monicasousa@esdrm.ipsantarem.pt (M.S.); af_martins17@hotmail.com (A.D.M.); jmoutao@esdrm.ipsantarem.pt (J.M.); salves@esdrm.ipsantarem.pt (S.A.); 2Life Quality Research Centre, 2040-413 Rio Maior, Portugal; 3Research Centre in Sport Sciences, Health Sciences and Human Development, 5001-801 Vila Real, Portugal; 4Comprehensive Health Research Centre (CHRC), Departamento de Desporto e Saúde, Escola de Saúde e Desenvolvimento Humano, Universidade de Évora, Largo dos Colegiais, 7000-727 Évora, Portugal

**Keywords:** fibromyalgia, exercise, multicomponent training, aerobic training, resistance training, strength training

## Abstract

Fibromyalgia is a rheumatic disease characterised by chronic widespread muscular pain and its treatment is carried out by pharmacological interventions. Physical exercise and a healthy lifestyle act as an important mechanism in reducing the symptoms of the disease. The aims of this study were to analyse and systematise the characteristics of combined training programs (i.e., type and duration of interventions, weekly frequency, duration and structure of training sessions and prescribed intensities) and to analyse their effects on people diagnosed with fibromyalgia. A systematic literature search was performed using the PRISMA method and then randomised controlled trial articles that met the eligibility criteria were selected. The Physiotherapy Evidence Database scale was used to assess the quality and risk of the studies. A total of 230 articles were selected, and in the end, 13 articles met the defined criteria. The results showed different exercise interventions such as: combined training, high-intensity interval training, Tai Chi, aerobic exercise, body balance and strength training. In general, the different interventions were beneficial for decreasing physical symptoms and improving physical fitness and functional capacity. In conclusion, a minimum duration of 14 weeks is recommended for better benefits. Moreover, combined training programs were the most effective for this population, in order to reduce the symptoms of the disease with a duration between 60 and 90 min, three times a week with a light to moderate intensity.

## 1. Introduction

Fibromyalgia (FM) is defined as a chronic rheumatic disease and is characterised by chronic widespread pain, muscle stiffness, sleep disturbances and cognitive problems [1,2,3,4]. In addition to these, the following symptoms are also observed: a feeling of fatigue and changes in the psychological state [5]. Moreover, FM can include muscle pain in the tender points, excessive fatigue, muscle strength loss and some psychological problems as mentioned before (i.e., sleep issues, anxiety, depression and reduced levels of satisfaction with life and self-esteem) [6,7]. Most of the time, the diagnosis is quite difficult to perform because there is no accurate (i.e., validated) diagnostic test to identify the disease. Thus, the diagnosis of this disease is carried out through palpation from tender points specific for FM [1]. 

Studies indicate that FM affects, on average, 2.1% of the world’s population and 2.31% of the European population, implying a painful loss of quality of life for the people who suffer from it and high economic costs [8]. The literature also points out that FM is more prevalent in women with values between 2.4% and 6.8% and in urban areas between 0.7% and 11.4% [9]. In Portugal, the prevalence is estimated at 1.7% (1.1% to 2.1%) [10]. 

Scientifically, the exact cause of the origin of FM remains unknown, so all the treatments of this disease are directed towards the reduction in the signs and symptoms presented [11]. In addition, the clinical control of the patient is carried out mainly through pharmacological interventions [12]. However, this type of treatment is not effective in solving functional problems, namely the loss of mobility and muscle strength and power, which negatively interferes with the quality of life of patients [13,14]. In this sense, some studies have demonstrated the importance of including non-pharmacological treatments in this pathology, mainly the regular practice of physical exercise associated with a healthy lifestyle [5,15].

Physical exercise promotes several benefits on a physical and psychological level. A physical exercise program works as an important mechanism that positively influences this population, attenuating the main symptoms, such as: the feeling of fatigue, depression, anxiety, muscle stiffness and sleep disturbances [16]. In this way, physical exercise has been used as a form of non-pharmacological intervention [17]. 

The American College of Sports Medicine (ACSM) recommends performing strength exercises (2 to 3 days/week), aerobic exercises (2 to 4 days/week) and flexibility exercises (1 to 3 days/week) to attenuate or reduce the signs and symptoms of FM [18]. In this sense, a combined training program may adjust to the recommendations for this population [19]. due to the fact that it involves aerobic, strength and stretching exercises, simultaneously, inducing several important adaptations in order to cover a greater number of symptoms. Consequently, strength, power, and aerobic capacity and power improvements may occur [20]. This type of training can be performed in the same session or in different sessions [21]. In this sense, aerobic exercise induces adaptations in various functional capacities such as transport, capture and the use of oxygen [22]; strength training becomes essential for increasing muscle strength [23,24]; and stretching exercises are beneficial to reduce the loss of mobility due to its constant immobilisation associated with pain [25]. 

To better understand the benefits of different types of exercises and physical therapy in FM, a set of studies were reviewed to obtain a comprehensive guideline for the prescription of exercise in this population [26]. The results suggest that individuals with FM have different responses to different types of exercise programs (e.g., aerobic training or strength training), since these same individuals present a great diversity of signs and symptoms. Accordingly, preference should be given to more global exercise protocols that are able to provide positive effects to the greatest number symptoms possible. In this way, it is important to better understand the effects of combined training and recommendations regarding the prescription of physical exercise in individuals diagnosed with FM.

In this way, the objectives of this systematic review were to analyse and systematise the characteristics of combined training programs (i.e., type and duration of interventions, weekly frequency, duration and structure of training sessions and prescribed intensities) and to analyse their effects on people diagnosed with FM.

## 2. Materials and Methods

This systematic review was performed following Preferred Reporting Items for Systematic Reviews and Meta-Analyses (PRISMA) 2020 [27] and the guidelines for performing systematic reviews in sports sciences [28]. The systematic review protocol was a priori registered in the OSF platform with the associated project number osf.io/v37s4.

### 2.1. Eligibility Criteria

The studies included in the present systematic review had the following inclusion criteria: (i) participants ≥ 18 years old with FM and autonomy, without other diseases (e.g., diabetes, hypertension and/or cardiovascular diseases); (ii) studies with combined training programs (aerobic and strength) with duration ≥4 weeks; (iii) exercise training programs supervised by a multidisciplinary team including a fitness exercise professional; (iv) randomised clinical trials; (v) studies written in English because it is the universal language.

The following items were considered the exclusion criteria: (i) participants < 18 years old with other diseases (e.g., diabetes, hypertension and/or cardiovascular diseases); (ii) studies with durations lower than 4 weeks and/or without combined training; (iii) studies written in other languages than English; (iv) other studies than randomised clinical trials.

### 2.2. Information Sources and Search Strategy

A systematic search of three databases (Web of Science, PubMed and EBSCO) was performed until 14 September 2022. Additionally, a manual search on the references of the included articles was also performed. 

The search strategy included the Boolean AND/OR and the following keywords: “fibromyalgia” AND “concurrent training” OR “combined training” OR “cross training”. The search strategy and their specificities from each database are presented in Table 1.

### 2.3. Selection and Data Collection Processes

All articles found by the aforementioned search strategy were evaluated by two authors (M.S. and A.D.M.) through titles and their abstracts in order to exclude duplicates and articles that did not meet the inclusion criteria. In addition, the abstracts that did not provide enough information were selected for a complete evaluation of the full article. In a second phase, the two authors evaluated all the articles in full to carry out a second selection according to the inclusion criteria. The lack of consensus between the two investigators was resolved in a meeting with the third investigator (R.O.). Then, M.S. and R.O. extracted the data while J.P.B reviewed the process.

### 2.4. Data Items

The following data were extracted from the selected articles: population characteristics such as sample size, sex, age, years of diagnosed FM, country, body mass index (BMI); intervention: characteristics of combined training programs (i.e., exercises and materials; weekly frequency and duration of training programs; intensity; sets and repetitions); outcomes: instruments/tools (type, manufacturer and questionnaires); aim and main results of the studies.

### 2.5. Study Risk-of-Bias Assessment

The Physiotherapy Evidence Database (PEDro) scale was applied to assess the risk of bias of the included studies. This PEDro scale was previously validated and its reliability confirmed [29]. The PEDro scale rates eleven criteria topics, in which 10 classify the overall score of the article, ranging from 0 (lowest quality) to 10 (highest quality). The classification of the scores was the following: “poor” (<4 points), “fair” (4–5 points), “good” (6–8 points) and “excellent” (9–10 points). Two authors (A.D.M. and R.O.) independently reviewed and rated the included articles, based on the PEDro scale. Then, the same authors shared the scores and discussed them on a point-by-point basis. When a consensus was not reached, a third author (J.P.B) was invited to its classification to make a final decision. 

### 2.6. Certainty Assessment

Based on the physiotherapy evidence database scale, Tulder et al.’s [30] criteria were applied to assess the interventions’ evidence. Thus, a study with a physiotherapy evidence database score of ≥6 is considered level 1 (high methodological quality) (6–8: good, 9–10: excellent) and a score of 5 or less is considered level 2 (low methodological quality) (4–5: moderate; <4: poor). 

Due to the clinical and statistical heterogeneity of the results, a qualitative review was performed, conducting a best-evidence synthesis [31,32]. This classification indicates that if the number of studies displaying the same level of evidence for the same outcome measure or equivalent is lower than 50% of the total number of studies, no evidence can be concluded regarding any of the methods involved in the study.

## 3. Results

### 3.1. Study Identification and Selection

A total of 335 articles were found across the three databases. All studies were exported using reference management software (EndNoteTM 20.0.1, Clarivate Analytics, Philadelphia, PA, USA). A total of 105 duplicate articles were recorded and subsequently removed. The remaining 230 articles were analysed by their titles and abstracts, and when insufficient information was available, the article was read in full, resulting in the removal of 288 articles deemed not to be in the scope on this review. Finally, after a complete reading of all articles, 34 more articles were excluded for not meeting the eligibility criteria. Thus, 13 articles were included in this systematic review (Figure 1).

### 3.2. Study Characteristics

The sample of articles selected for this systematic review included 13 studies published between 2000 and 2020. The studies covered an adult population diagnosed with FM and females with ages ranging from 30 to 59 years. There were different exercise interventions such as combined training, high-intensity interval training, Tai Chi, aerobic exercise, body balance and strength training. Moreover, several instruments/tests were used to assess pain, sleep quality, health status and strength gains in the upper and lower limbs. The characteristics of the articles included in the systematic review are presented in Table 2.

### 3.3. Risk of Bias in Studies

Table 3 presents the assessment of risk of bias (PEDro scale). The criteria with lower scores were related to the blinding of all participants and blinding of all persons who administered the training protocols. Moreover, no study was classified with poor methodological quality. 

### 3.4. Intervention Characteristics

The characteristics of the interventions exercise programs are presented in Table 4. Regarding the exercises included, only main phases of each training have been reported in the table.

## 4. Discussion

This systematic review aimed to analyse and systematise the characteristics of combined training programs and their effects in individuals diagnosed with Fibromyalgia. In the studies that were analysed, significant values were found for at least one of the evaluated parameters in all studies: (i) physical fitness tests [1,23,33,36,37,39,40,41]; (ii) decreased symptoms and impact of FM on participants [1,5,15,17,23,33,36,37,41]; (iii) lower limb strength [1,15,33]. These results are in line with other authors who claim that physical exercise programs are important stimuli with positive influence, attenuating the symptoms of the disease, through changes in the hypothalamic–pituitary–adrenal axis (HPA)—resulting in the release of neurotransmitters due to exercise and controlling and/or reducing localised pain [16,42]. The major findings were the improvement of health-related life quality, pain intensity, stiffness, fatigue, physical function, withdrawals and absence of adverse events [17,43,44,45]. It is relevant to point out that the thirteen studies in the present review provided moderate- to good-quality evidence for the mental dimension of the SF-36, VAS, FIQ, IPAQ, improvement in %fat, FM, BMI, 6-MWT and decrease in BDI-II, pain, fatigue and sleep. For instance, Bidonde et al. [43], who used a 15% threshold for calculation of the clinically relevant differences between experimental and control groups, reported that eight trials provided low-quality evidence for pain intensity, fatigue, stiffness and physical function, and moderate-quality evidence for withdrawals and HRQL at completion of the intervention (6 to 24 weeks).

Regarding exercise programs, previous research has shown that combined exercise programs have more positive effects compared to single-type exercises in FM [5,46]. Both the aerobic, strength and combined training protocols showed positive effects on the participants as also reported by Bidonde et al. [43]. Results showed that strength training, aerobic training and combined exercise programs resulted in favourable effects on FM symptoms [1,5,15,23,33,35,37,40,41]. Among these, the aerobic and combined interventions presented the highest effects on reducing the FIQ. Flexibility interventions, however, were not significant for reducing it [47,48,49]. Exercise programs lasting between 13 and 24 weeks and training sessions lasting no more than 60 min seemed to be associated with greater improvements in pain relief. Regarding the durations of the exercise programs, it seems that the programs longer than 6 weeks have more positive effects in this population. The association of aerobic exercises and Tai Chi also revealed positive effects on the symptoms presented for this population [5,23]. Aerobic exercises seem to be fundamental stimuli in exercise programs as they induce adaptations in several systems, namely in the cardiovascular, energetic, neuromuscular and neuroendocrine systems [43]. The latter allows an increase in serotonin and norepinephrine concentrations, with a consequent improvement in mood and greater physical well-being [16,47]. The norepinephrine and serotonin are involved in the modulation of arousal and mood and have been related to a variety of affective functions as well as associated clinical dysfunctions [50,51]. The norepinephrine modulates drive and energy and exerts a fine regulation of specific processes including learning, memory, sleep, arousal and adaptation [52]. Further, exercise appears to reduce serotonin transporter expression, increase serotonin levels and increase opioid levels in central inhibitory pathways, suggesting that exercise can reduce pain by utilising our endogenous inhibitory systems [50,51,52].

Regarding studies that evaluated the effects of aerobic exercise combined with Tai Chi exercises, it was found that participants who performed Tai Chi exercises improved in all assessment parameters, such as: impact of Fibromyalgia (FIQ); pain threshold; anxiety and depression (Hospital Anxiety and Depression Scale and BDI-II); and finally, sleep quality (Pittsburgh sleep quality index) compared to participants who performed only aerobic exercise, who had less improvement. It should also be noted that the combined Tai Chi and aerobic exercise groups improved the impact of Fibromyalgia (FIQ) more in the 24-week Tai Chi groups than in the 12-week group [17,36]. In addition to these parameters, there were also enhancements in functional capacity tests, such as sit and reach, and in the 6-MWT, in terms of depression and anxiety [36]. Regarding the type of exercise programs in order to reduce the symptoms of the disease, it appears that combined training programs are the most effective for this population.

Regarding the effects of exercise programs on muscle strength, the studies analysed in this systematic review found that the combined exercise of aerobics and strength promotes higher improvements in strength of concentric extension in the legs compared to the groups that performed only aerobic exercise or strength training [23,34,35,40]. Exercise programs including strength training, endurance and aerobics are accepted as a standard treatment protocol in FM [33,46,53,54]. There is growing evidence and a strong recommendation of using aerobic and strength training together for FM treatment [55].

The studies present different types of exercise, such as aerobics, strength, flexibility and combined interventions, but some studies point to the beneficial effects of using aquatic exercises. The properties of water and the physical activity performed in warm water seem to positively affect FM symptoms [39,41]. Consequently, these physiological changes can help to relax the muscles of the body [56]. In view of the benefits, the practice of physical exercises in water has been indicated to reduce symptoms such as pain, anxiety and depression [1].

As evidenced in the literature, the main symptoms of the disease are pain and constant immobilisation due to this fact [57]. As such, participants, when immobilised in their daily lives, lose muscle strength, which can lead to their disability and reduced quality of life [58]. Therefore, muscle strengthening exercises are essential for gaining muscle mass to generate the strength needed for daily tasks in this population. Nevertheless, it was also verified that a training protocol lasting up to 6 weeks did not generate greater benefits in relation to the evaluated parameters, such as the pain threshold, the impact of FM (FIQ), the functional capacity and the muscle strength gains that were greater between 14 and 24 weeks [15,34,39]. This evidence can be justified by the fact that this type of program is carried out with light to moderate intensity, due to the type of pathology and, as such, the results are not immediately observable, rather in the long term [52]. In addition to these parameters, the distance covered (6-MWT) improved significantly for participants in the exercise groups [14,33,40]. It is relevant to highlight that according to the ACSM [18], light aerobic activity is considered as <40% and moderate is 40 to 59% of the HR reserve. 

However, in the study by Atan and Karavelioglu [1], high-intensity interval training (HIIT) plus strengthening and stretching exercises and moderate-intensity continuous training (MICT) plus strengthening and stretching exercises interventions showed significant improvements for FM effect, pain degree, functional capacity and quality of life compared to the control group. HIIT was not superior to MICT, but body composition parameters improved significantly only for the MICT group. It is important to draw attention to the fact that this study was the only one to use an intensity higher than 59% of the HR reserve and, even so, some improvements were not noted. 

Light to moderate intensity programs are recommended for this type of population, which could be a reason why the HIIT group has not achieved improvements in pain symptoms [59]. However, it should be noted that when muscle strengthening exercises are combined with flexibility exercises, improvements were observed in the pain threshold, as well as in the participants’ quality of life [34,35]. These improvements are justified by the fact that flexibility exercises are used as a way to increase the range of motion of one or more joints, reinforcing once again that these participants lose their mobility due to their constant mobilisation associated with pain [60]. An important point to consider is related to the intensity of strength training, since the protocols were not revealed. One study used six sets [30], while the other used just one set [31]. Even so, the number of repetitions followed the ACSM guidelines [18,61]. 

Studies that evaluated the effects of aerobic exercise combined with a physical exercise program for muscle strengthening and flexibility and the group that performed only supervised aerobic exercise demonstrated improvements in depression and in the quality of life associated with health in both groups. However, beneficial improvements in quality of life (SF-36) were observed in both exercise groups, with the supervised aerobic exercise group demonstrating improvements in the dimensions of physical and social functioning, while the supervised aerobic exercise group combined with muscle strengthening and flexibility exercises demonstrated improvements in the dimensions of physical functioning, body pain, vitality and mental health [5].

This study reviews the effects of combined training programs in individuals with FM, summarising recent reviews and describing new advances in the research related to progressive exercise regimens (aerobic, strength and flexibility interventions), and other forms of physical activity applied to FM (e.g., Tai Chi, Yoga and Pilates). However, since the population with FM presents heterogeneity, including different years of diagnosis, further investigation with regard to the short- and long-term response patterns to exercise and physical activity prescription may lead to a better customisation program in order to improve exercise adherence and optimise the benefits of exercise and physical activity. 

For instance, six studies did not present the years of diagnosis [15,23,35,36,37,38,39], while the remaining studies ranged from 1 to 13.8 years [1,17,33,34,40,41]. Such information should be considered by fitness professional and exercise physiologists when interpreting the positive effects. 

Thus, future research examining the effects of exercise and physical activity for people with FM are needed to elucidate the best dose–response curve for exercise intensity, frequency and duration on symptoms. The assessment of the long-term effects on health in this population and how long the positive effects are sustained should also be studied. Another aspect to consider in future studies is the race description of participants. From the 13 studies included, only one described that information [15]. Finally, when conducting a future systematic review study on this population, a meta-analysis would be interesting to strengthen the results of the same intervention type, which was not possible in the present study.

## 5. Conclusions

According to the studies included in this systematic review, it was concluded that the practice of physical exercise is, in general, beneficial and essential for improving the symptoms of the disease, physical fitness and functional capacity and also in terms of anxiety and depression.

Through the analysed studies, the most appropriate training protocol for women diagnosed with FM, with the aim of mitigating the various symptoms of the disease, should contain the following parameters:(i)Minimum duration of 14 weeks and never less than 6 weeks. Programs lasting 6 weeks did not have such positive effects in terms of the symptoms of the disease.(ii)Combined training programs are the most effective for this population, in order to reduce the symptoms of the disease. Training programs composed of aerobic exercises, strength training and stretching are the most indicated.(iii)Duration of sessions between 60 and 90 min, with the objective of executing the program outlined according to the limitations of each participant.(iv)Carry out the exercise program at least up to 3 times a week.(v)Aerobic exercises should be performed at 60–65% HRmax.(vi)Perform one set of exercises for large muscle groups (associated with pain points), consisting of 8 exercises and performing 8–10 repetitions in an initial phase, progressing to 15 repetitions. Rest at least 1–2 min between exercises.(vii)Perform static stretching exercises lasting 30–60 s for pain points.(viii)The intensity of the program should be light to moderate, following the ACSM guidelines for aerobic exercise.

The results of the present systematic review demonstrate that exercise programs that include aerobic exercises, strength training and stretching exercises are the most beneficial to reduce the symptoms of the disease. Aerobic exercises are essential for the transport and use of oxygen; strength training is essential for gaining muscle mass, generating strength for the patients’ daily tasks; and stretching is essential for their mobility. 

## 6. Future Lines of Research

For future studies, a follow-up of the program is recommended. It would be interesting to evaluate the participants over a year, where several assessments would be carried out at different times regarding their physical fitness and the impact of FM. In this way, it would be possible to collect larger sample data for a better comparison between moments. 

## Figures and Tables

**Figure 1 healthcare-11-01708-f001:**
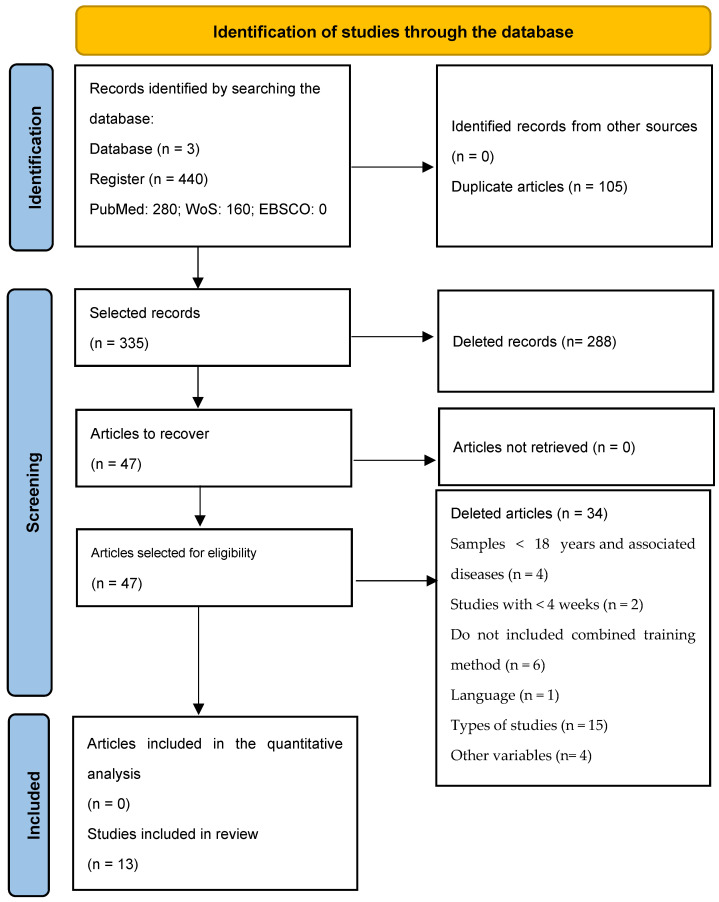
Flowchart of the Systematic Review.

**Table 1 healthcare-11-01708-t001:** The complete search strategy for each database.

Database	Specificities of the Databases	Search Strategy	Number of Articles in Automatic Research
PubMed	Search for title and abstract also includes keywords	(“fibromyalgia”) AND (“concurrent training” OR “combined training” OR “cross training”	280
Web of Science	Search for title and abstract also includes keywords	(“fibromyalgia”) AND (“concurrent training” OR “combined training” OR “cross training”	160
EBSCO	Search for title and abstract also includes keywords	(“fibromyalgia”) AND (“concurrent training” OR “combined training” OR “cross training”	0

**Table 2 healthcare-11-01708-t002:** Characteristics of the articles included in the systematic review.

Author (Year)	Country	Objectives	Participants by Gender (N)	Age (M ± SD)	Years of Diagnosis	Instruments/Tests/Evaluation Tools and Variables
Gulsen et al., 2020 [33]	Turkey	To evaluate the effects of training combined with immersive virtual reality	N = 16; EG = 8 EG + Immersive Virtual Reality = 8	EG = 38.5 (29.5–50.0) EG + Immersive Virtual Reality = 46.5 (36.5–49.5)	EG = 4 (2–7.5) EG + Immersive Virtual Reality = 4 (2.5–8)	VAS: Pain Biodex Balance System (Shirley, NY, USA): Balance FIQ: Impact of FM Questionnaire IPAQ: Levels of PA Questionnaire 6-MWT: Aerobic Capacity SF-36: Health status of population Questionnaire
Atan and Karavelioglu et al., 2020 [1]	Turkey	To compare high-intensity interval training versus a combined training of continuous moderate intensity and strength plus stretching exercises	N = 45 HIIT Group = 19 MICT Group = 19 CG = 17	HIIT Group = 46.5 ± 9.4 MICT Group = 47.3 ± 8.0 CG = 52.7 ± 8.9	HIIT Group = 3.1 MICT Group = 2.0 CG = 2.3	FIQ VAS: Pain SF-36: Health status of population Questionnaire Maximal Cardiopulmonary Exercise Test InBody 720, Biospace: Body composition (Weight, waist circumference and BMI)
Wang et al., 2018 [17]	United States of America	To evaluate the effects of Tai Chi protocol versus aerobic exercise	N = 226 Tai Chi Group = 151 1 × 12 weeks = 39 2 × 12 weeks = 37 1 × 24 weeks = 39 2 × 24 weeks = 36 AEG 2 × 24 weeks = 75	Tai Chi Group: 1 × 12 weeks = 53 ± 12.6; 2 × 12 weeks = 52,1 ± 10.3; 1 × 24 weeks = 50.8 ± 11.8; 2 × 24weeks = 52.1 ± 13.3 AEG 2 × 24 weeks = 50.9 ± 12.5	Tai Chi Group: 1 × 12 weeks = 11.1 ± 8.6; 2 × 12 weeks = 12.6 ± 12.1; 1 × 24 weeks = 12 ± 8.3; 2 × 24 weeks = 13.8 ± 10.4; AEG 2 × 24 weeks M = 11.3 ± 8.7	FIQ VAS: Pain Depression and Anxiety Questionnaire Pittsburgh Sleep Quality Questionnaire BDI-II: Depression and behavioural manifestations Questionnaire
Celenay et al., 2017 [15]	Germany	To compare the effectiveness of a 6-week combined exercise program with and without CMT on pain, fatigue, sleep problems, health status and quality of life	N = 20 EG N = 20 EG + connective tissue massage = 20	EG = 39.9 ± 9.5 $\bar{x}$ = 42.5 ± 8.3	ND	IPAQ-7: Levels of PA Questionnaire VAS: Pain Sleep: Quality of sleep Questionnaire FIQ: Impact of FM Questionnaire SF-36: Health Status population Questionnaire
Sañudo et al., 2013 [34]	Spain	To determine the effect of body balance and dynamic strength of an exercise program complemented with WBV	N = 46 EG + WBV (WBVEX) = 15 EG = 15 CG = 16	EG + WBV = 57.1 ± 6.8 EG = 62.2 ± 9.8 CG = 55.5 ± 7.9	EG + WBV = 8.5 ± 7.4 EG = 9.2 ± 8.3 CG = 8.8 ± 8.2	Biodex F1C Stability System (BSS; Biodex, Inc. Shirley, NY, USA): Body Balance and Lower Limb Dynamic Strength
Sañudo et al., 2012 [35]	Spain	To analyse the effects of balance and strength through an exercise training program combined with WBV	N = 30 EG + WBV = 15 EG = 15 CG = 16	$\bar{x}$ = 59 ± 7.9	ND	Biodex Stability System (BSS, Biodex, Inc., Shirley, NY): Body Balance The Galileo Fitness Platform (Novotech, Germany): Evaluation of knee extensor muscle strength
Romero-Zurita et al., 2012 [36]	Spain	To analyse the effects of Tai Chi training in women	N = 23	$\bar{x}$ = 51.4 *±* 6.8	ND	Body composition and anthropometric measurements: Weight, Waist Circumference, BMI FIQ SF-36: Health Status population Questionnaire Depression and Anxiety Questionnaire Vanderbiet Pain Management Inventory: Copping strategies Rosenbery Self-Esteem Scale: global self esteem General Self-Efficacy Scale: Beliefs in her/his own capabilities to attain aims
Sañudo et al., 2011 [37]	Spain	To analyse the effects on perceived health status, functional capacity and depression of a long-term exercise program versus usual care	N = 42 EG = 21 Usual Care CG = 21	EG = 55.4 *±* 7.1 Usual Care CG = 56.1 *±* 8.4	ND	FIQ SF-36: Health Status Population Questionnaire BDI: Attitudes and symptoms of stress Questionnaire
Sañudo et al., 2010 [38]	Spain	To determine the effects of supervised aerobic exercise and a supervised exercise program combined with aerobic exercise, strength and flexibility	N = 64 AEG2 CTG = 21 CG N = 21	AE group = 55.9 *±* 1.6; CTG M = 55.9 *±* 1.7; CG = 29.7 *±* 1.1	ND	FIQ SF-36: Health Status Population Questionnaire BDI: Attitudes and symptoms of stress Questionnaire 6-MWT: Aerobic Capacity Hand-grip strength: Measure of muscular strength or the maximum force/tension by forearm muscles Flexion and extension (shoulders and hips): degrees
Carbonell-Balza et al., 2010 [39]	Spain	To analyse the effects on pain, body composition and physical fitness of a multidisciplinary intervention in women.	N = 75 EG = 41 CG = 34	EG = 50 *±* 7.3 CG = 51.4 *±* 7.3	ND	InBody 720; Biospace, Gateshead, UK: Body fat and muscle mass Functional Fitness Test Battery: lower and upper body strength and flexibility
Valkeineu et al., 2008 [23]	Finland	To determine the effects on muscle strength, aerobic and functional performance on postmenopausal symptoms of a combined strength and resistance training in women	N = 26 EG = 15 CG = 11	EG = 59 *±* 3 CG = 58 3	ND	Health Assessment Questionnaire: Self-report functional status (disability) measures VO2 peak: Maximum oxygen carrying capacity with a bicycle ergometer test
King et al., 2002 [40]	Canada	To examine the effectiveness of a supervised aerobic exercise program, a self-management education program, and an exercise and education program for women	N = 152 EG = 46 Education group = 48 Exercise and Education Group = 37 CG = 39	EG = 45.2 *±* 9.4 Education group = 44.9 *±* 10.0 Exercise and Education Group = 47.4 *±* 9.0 CG = 47.3 *±* 7.3	EG= 7.8 *±* 6.1 Education group= 10.9 *±* 10.7 Exercise and Education Group = 8.9 *±* 7.3 CG = 9.6 *±* 7.9	FIQ 6-MWT: Aerobic Capacity
Mannerkorpi et al., 2000 [41]	Sweden	To determine the effects of a pool-based exercise training program combined with an education program	N = 69 EG = 37 CG = 32	EG = 45 *±* 8.0 CG = 47 *±* 11.6	EG = 8.9 *±* 7.2 CG = 8.4 *±* 6.0	FIQ 6 MW: Aerobic Capacity SF-36: Health Status population Questionnaire Arthritis Self Efficacy Scales: Pain and activities of daily living Questionnaire Arthritis Impact Measures Scales: Weight-bearing, posture and antigravity movement Questionnaire Quality of Life Questionnaire: Individual’s physical, psychological and social well-being Questionnaire

$\bar{x}$: mean; RCT: Randomised Controlled Trial; VAS: Visual Analog Scale; FIQ: Fibromyalgia Impact Questionnaire; IPAQ: International Physical Activity Questionnaires; 6-MWT: 6-Minute Walk Test; SF-36: Short-Form 36; BDI-II: Beck Depression Inventory-II; BMI: body mass index; FM: Fibromyalgia; PA: Physical Activity; ND: Non-Described; EG, Exercise Group; AEG, Aerobic Group; CTG, Combined training Group; HIIT Group, High-Intensity Interval Training Group; MICT, moderate-intensity continuous training; CG, Control Group; WBV: Whole-Body Vibration; WBVEX: Whole-Body Vibration Exercise Group.

**Table 3 healthcare-11-01708-t003:** Risk of bias assessment (PEDro scale).

Study	PEDro Scale											Total Score	Methodological Quality
	C1	C2	C3	C4	C5	C6	C7	C8	C9	C10	C11		
Gulsen et al. [33]	1	1	0	1	0	0	1	0	0	1	1	5	Moderate
Atan and Karavelioglu et al. [1]	1	1	1	1	0	0	1	1	0	1	1	7	Good
Wang et al. [17]	1	1	1	1	0	0	1	0	1	1	1	7	Good
Celenay et al. [15]	1	1	1	1	0	0	0	0	0	1	1	5	Moderate
Sañudo et al. [34]	1	1	1	1	0	0	0	1	1	1	1	7	Good
Sañudo et e al. [35]	1	1	1	1	0	0	0	1	0	1	1	6	Good
Romero-Zurita et al. [36]	1	1	1	1	0	0	0	1	1	1	1	8	Good
Sañudo et al. [37]	1	1	1	1	0	0	0	1	1	1	1	8	Good
Sañudo et al. [38]	1	1	1	0	0	0	0	1	1	1	1	6	Good
Carbonell-Balza et al. [39]	1	1	1	1	1	0	0	1	1	1	1	9	Excellent
Valkeineu et al. [23]	1	1	0	1	0	0	1	1	0	1	1	6	Good
King et al. [40]	1	1	0	1	0	0	1	0	1	1	1	6	Good
Mannerkorpi et al. [41]	1	1	0	1	0	0	1	0	0	1	1	5	moderate

C1: eligibility criteria were specified; C2: participants were randomly allocated to groups; C3: allocation was concealed; C4: the groups were similar at baseline regarding the most important prognostic indicators; C5: there was blinding of all participants; C6: there was blinding of all therapists who administered the therapy; C7: there was blinding of all assessors who measured at least one key outcome; C8: measures of at least one key outcome were obtained from more than 85% of the participants initially allocated to groups; C9: all participants for whom outcome measures were available received the treatment or control condition as allocated, or, where this was not the case, data for at least one key outcome were analysed according to “intention to treat”; C10: the results of between-group statistical comparisons are reported for at least one key outcome; C11: the study provides both point measures and measures of variability for at least one key outcome. Note: C1 values do not count for the total score.

**Table 4 healthcare-11-01708-t004:** Characteristics of the Exercise Programs Present in the Systematic Review.

Author (Year)	Exercises	Frequency Program Length and Duration of Sessions	Intensity	Sets (N); Reps (N); Rest	Results
Gulsen et al., 2020 [33]	AEG: Treadmill; Pilates and IVR	Frequency: 2 × week Program length: 8 weeks Session duration: 80 min	AE = 60–80% HRmax	Non-Described	After the intervention, there were significant improvements in the exercise and Immersive Virtual Reality groups in pain, balance, impact of FM, fatigue, level of PA, functional exercise capacity and quality of life (*p* < 0.05). Exercise + IVR groups showed more improvements than exercise group in pain, fatigue, PA level, mental component and quality of life (*p* < 0.05).
Atan and Karavelioglu et al., 2020 [1]	HIIT—Cycle Ergometer MICT—Shoulder press with dumbbells or on machine; shoulder raises; bicep curl; squats; standing hip flexion and extension.	Frequency: 5 × week Program length: 6 weeks Time of the HIIT session— 35 min; MICT—55 min Control Group—without exercise	HIIT: 4 min sets at 80–95% of peak HR interspersed with three 3 min of active recovery intervals at 70% of peak HR and 5 min of return to calm at 50% of peak HR; MICT: 45 min of ST performed for the main muscle groups.	MICT: 1 set, 8–10 reps; HIIT: 4 sets of 4 min with 3 sets of 3 min of active recovery intervals and 5 min cool down period cycling	Group-time interactions were significant for the FIQ between interventions and control (*p* < 0.001). There were significant group-time interactions for the pain, SF-36 and cardiopulmonary exercise test parameters between treatments and control (all, *p* < 0.05). Body weight, fat percentage, fat-mass and BMI improved significantly (all, *p* < 0.05) only in MICT group after treatment.
Wang et al., 2018 [15]	Tai Chi: choreographed AE	Frequency: 1 or 2× week (Tai Chi and EA) Program: 52 weeks Session duration: 60 min	AE: 20 min of AE 50–60% HRmax and 11–13 on the RPE. From the 10th session, AE progressed to 60–70% of HRmax	Non-Described	Tai Chi groups improved significantly more than AEG in FIQ, anxiety and self-efficacy at 24 weeks either training 1 or 2 times per week (all, *p* < 0.05). Tai Chi groups compared with AE administered with the same intensity and duration (24 weeks, 2 times per week group) had greater benefits in FIQ *p* < 0.001. The groups who received Tai Chi for 24 weeks showed greater improvements in FIQ than Tai Chi Group of 12 weeks (*p* = 0.007).
Celenay et al., 2017 [15]	AE: walk on the treadmill for 20 min; ST: Deep neck muscles; deltoid; latissimus dorsi; pectoralis; scapular retractor muscles; external rotators of the shoulder; erector spinae; abdominals; and gluteus muscles.	Frequency: 2 × week Program duration: 6 weeks Session duration: 60 min	AE: 65–70% HRmax and then 75–80% HRmax; ST: Light/medium band and progress to a strong band	1 set ST: 10 reps with a progression to 15 reps;	In the Exercise + connective tissue massage group, pain, fatigue and sleep problems decreased; health status and quality of life improved (*p* < 0.05). Exercises with connective tissue massage were superior in improving pain, fatigue, sleep problems and role limitations due to physical health compared to exercise alone. General health perceptions parameters related to quality of life improved in the Exercise group than in the connective tissue massage group (*p* < 0.05).
Sañudo et al., 2013 [34]	Exercise Group: 10–15 min of AE. This is followed by 15–20 min of ST. Exercise Group + WBV: stood on the platform on both legs, with both knees in isometric 120° flexion. Exercises such as unilateral static squats were used.		AE: 65–70% HRmax WBV: Vibration frequency of 30 Hz and at a peak-to-peak displacement of 4 mm (71.1 m/s^−2^ ≈ 7.2 g).	ST: 1 set; 8–10 reps. WBV: 6 sets in the exercises in which the participants stand on the platform and 4 sets in the isometric 30 s exercises with 45 s of recovery.	There were no between-group differences in any outcome measures (all *p*-values > 0.05), except for MLMD with open eyes between both experimental groups (*p* = 0.02). The 8-week intervention of exercises and WBV resulted in a statistically significant improvement in MLS I, and significant differences for the WBVEX over the EX group (*p* = 0.014) and over the CG (*p* = 0.029).
Sañudo et e al., 2012 [35]	Exercise Group: combination of AE and ST. 10–15 min of AE; 15–20 min of ST Galileo Fitness_platform: stand up with both knees in isometric flexion plus unilateral static squats.	Frequency: 2 × week + 3 additional WBV sessions Program duration: 6 weeks Session duration: 45 min	Exercise Group: AE 65–70% HRmaxGalileo Fitness_platform: frequency of 20 Hz and variable amplitude of 2–3 mm.	ST: 1 set; 8 exercises; 8–10 reps; Galileo Fitness_platform: 3 sets of 45 s with a recovery of 120 s between sets and 4 sets of 15 s. The participants completed 15 s of the exercise on the right leg and then immediately completed 15 s on the left leg, and this was considered a set.	Exercise Groups of Medio-Lateral Stability Index improved balance when participants were assessed with eyes open and closed (all, *p* < 0.05).
Romero-Zurita et al., 2012 [36]	Yang style Tai-Chi forms,	Frequency: 3 × week Program: 28 weeks Session duration: 60 min	The average RPE value was 11 ± 1.	8 forms of Tai-Chi, Yang style	Patients showed improvements in pain threshold and total number of tender points (all, *p* < 0.001). Tai-Chi group improved the FIQ total score (*p* < 0.001) and six subscales: stiffness (*p* = 0.005), pain, fatigue, morning tiredness, anxiety and depression (all, *p* < 0.001). The intervention was also effective in six SF-36 subscales: bodily pain (*p* = 0.003), vitality (*p* = 0.018), physical functioning, physical role, general health and mental health (all, *p* < 0.001)
Sañudo et al., 2011 [37]	Exercise Group: muscle-strengthening exercises, consisting of a circuit of 8 exercise stations (shoulder press; shoulder raises; biceps curl; squats; hip flexion and extension; and standing abductors).	Frequency: 2 × week Program: 24 weeks Session duration: 60 min	AE: 65–70% HRmax	ST: 1 set, at each station 8–10 reps with dumbbells 1–3 kg;	Improvements in the Medio–Lateral Stability Index and Medio–Lateral Mean Deflection with open eyes were found in the whole-body vibration exercise group compared with the control group (*p* = 0.02).
Sañudo et al., 2010 [38]	AE: 15–20 min continuous walking with arm movements, aerobic dancing and jogging Combined exercise: 10–15 min AE; jogging; 15–20 min of muscle strengthening for 8 muscle groups: deltoids, biceps, neck, hips, back and chest.	Frequency: 2 × week Program: 24 weeks Session duration: 60 min	AEG: between 60% and 65% HRmax Combined Exercise: AE 65–70% HRmax	EA Group: 1 series for muscle strengthening and flexibility exercises. 6 exercises of 1.5 min of AE Combined Exercise: 1 series for muscle strengthening and flexibility exercises. Muscle strengthening consisting of 8–10 reps with a load between 1 and 3 kg and flexibility: 3 reps for 8–9 exercises, holding the static position for 30 s.	An improvement from baseline in total FIQ score was observed in the exercise groups (*p* < 0.002) and it was accompanied by decreases in BDI scores of 8.5 (*p* < 0.001) and 6.4 (*p* < 0.001) points in the AE and CE groups, respectively. Relative to non-exercising controls, CE evoked improvements in the SF-36 physical functioning (*p* = 0.003) and bodily pain (*p* = 0.003) domains and it was more effective than AE for evoking improvements in the vitality (*p* = 0.002) and mental health (*p* = 0.04) domains. Greater improvements were observed in shoulder/hip range of motion and handgrip strength in the CE group.
Carbonell-Balza et al., 2010 [39]	1st session, pool resistance exercises developed at a slow pace using water and aquatic materials as a means of resistance; 2nd session, pool-balance-oriented activities: position changes, walking backwards, coordination through aquatic exercises and dance exercises; 3rd session: land-based AE and coordination through an exercise circuit and 90 min of psychological follow-up.	Frequency: 3 × week Program: 12 weeks Session duration: 45 min of physical exercise classes; 90 min of psychological support	The average RPE was 12–13 AU.	Non-Described	A significant groupxtime effect for the left (L) and right (R) side of the anterior cervical (*p* < 0.001) and the lateral epicondyle R (*p* = 0.001) tender point. Pain threshold increased in the intervention group (positive) in the anterior cervical R (*p* < 0.001) and L (*p* = 0.012), and in the lateral epicondyle R (*p* = 0.010), whereas it decreased (negative) in the anterior cervical R (*p* < 0.001) and L (*p* = 0.002) in the usual care group.
Valkeineu et al., 2008 [23]	Concurrent training: in the 1st week, participants performed 2 strength training sessions and 1 endurance training session and in the 2nd week, 1 strength training and 2 endurance training sessions, and vice versa on alternate weeks. Strength training included isometric leg extension, concentric leg extension, elbow flexion and trunk flexion and extension.	Frequency: 3 × week Program: 21 weeks of combined training (strength and endurance training) Session duration: 60–90 min	Bicycle: 50 W with a load progression of 20 W until exhaustion Strength: 1 RM.	Bicycle: 3 min of heating with an intensity of 50 W and the load was increased to 20 W until exhaustion ST: 3 sets of exercises for legs, hips and knees and elbow flexors. Re-accelerate to maximum force for 3 to 5 s/1 min rest between exercises.	The concurrent training showed higher values in Wmax (*p = 0*.001), work time (*p* = 0.001), concentric leg extension force (*p* = 0.043), walking (*p* = 0.001), stair-climbing (*p* < 0.001) time and fatigue (*p* = 0.038) than strength training. The training led to an increase of 10% (*p* = 0.004) in Wmax and 13% (*p* = 0.004) than control group.
King et al., 2002 [40]	Education Group was based on self-management principles, where information was given about FM and individual goals and strategies for dealing with pain or other symptoms were established, guiding the participants toward a balanced and active life. Exercise Group included AE walking in deep or shallow water or low-impact AE such as walking outdoors. Education + EG was a combination of the previous protocols.	Frequency: 3 × week Program: 12 weeks Session duration: Exercise Group: 40 min Education Group: 60 min	AE: 60–75% HRmax	Non-Described	Only Exercise Group showed higher distance in the Six-Minute-Walk test (*p* = 0.04) when compared with Education Group.
Mannerkorpi et al., 2000 [41]	Exercise program in a heated pool and included resistance, flexibility, coordination and relaxation exercises.	Frequency: 1 × week Program: 24 weeks Session duration: 35 min	ND	Non-Described	Significant differences between the treatment group and the control group were found for the FIQ total score (*p* = 0.017) and the 6 min walk test (*p* < 0.001). Significant differences were also found for physical function, grip strength, pain severity, social functioning, psychological distress and quality of life for the treatment group.

VAS: Visual Analog Scale; FIQ: Fibromyalgia Impact Questionnaire; IPAQ: International Physical Activity Questionnaires; 6-MWT: 6-Minute-Walk Test; SF-36: Short-Form 36; CPET: Cardiopulmonary Exercise Test; BDI-II: Beck Depression Inventory-II; BMI: body mass index; PA: Physical Activity; EG, Exercise Group; AEG, Aerobic Group; CTG, Combined Training Group; HIIT Group, High-Intensity Interval Training Group; MICT, moderate-intensity continuous training; CG, Control Group; HRmax: heart rate maximal; RPE: rating of perceived exertion; AE: aerobic exercise; ST: strength training.

## Data Availability

Not applicable.

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
