# Peer review of "Effects of Combined Training Programs in Individuals with Fibromyalgia: A Systematic Review"

_healthcare, 2023, doi:10.3390/healthcare11121708_

Round 1
Reviewer 1 Report
The systematic review is interesting and will gather interest. I have few comments. 1. Indroduction section, the first paragraph is lack of direction. Please revise giving an overview of the study. 2. The search strategy should be described in detail. 3. Also add definition of "combined exercises". 4. Why were many articles in which only had a single-type exercise were also included in the tables? 5. Line166, "...resulting in the removal of 288 articles." Why? List detail reasons for exclusion. 6. Too many uncommon abbreviations are in the tables, making it extremely difficult to read them. 7. Line 270-271, why wasn't the 42nd reference included in the table? 8. Line 283-284, "...an increase in serotonin and norepinephrine concentrations...". The physiological effects of the two hormones are usually opposite, so how would they improve mood? Elaborate. 9. The logic in the discussion section is rather hard to follow and it is difficult to understand how the conclusions were reasoned out. Need to adda clear synthesis after discussing each point.Author Response
Reviewer 1
Comments and Suggestions for Authors
The systematic review is interesting and will gather interest. I have few comments.
- Introduction section, the first paragraph is lack of direction. Please revise giving an overview of the study.
Authors: Thank you for your suggestion. We revised the first paragraph accordingly.
- The search strategy should be described in detail.
Authors: Dear reviewer, information has been added in order to detail the search strategy.
- Also add definition of "combined exercises".
Authors: Dear reviewer, the revised version of the manuscript does not include any passage of “combined exercises”. Instead, the authors of this study used the expression of “combined training”. This is defined in introduction as follows: “In this sense, a combined training program may adjust to the recommendations for this population [17] due to the fact that it involves aerobic, strength and stretching exercises, simultaneously, inducing several important Deadaptations in order to cover a greater number of symptoms. Consequently, strength, power, and aerobic capacity and power improvements may occur [18]. This type of training can be performed in the same session or in different sessions [19].”
- Why were many articles in which only had a single-type exercise were also included in the tables?
Authors: Dear reviewer, those articles were included because they included exercise training programs
- Line166, "...resulting in the removal of 288 articles." Why? List detail reasons for exclusion.
Authors: Those articles were removed because the title and abstracts were not in the scope on this review.
- Too many uncommon abbreviations are in the tables, making it extremely difficult to read them.
Authors: Thank you for raising this point. We reduce the number of the uncommon abbreviations.
- Line 270-271, why wasn't the 42nd reference included in the table?
Authors: Dear reviewer, the previous reference 42 was a review study:
Okifuji, A.; Gao, J.; Bokat, C.; Hare, B.D. Management of Fibromyalgia Syndrome in 2016. Pain Manag 2016, 6, 383–400
Review studies were not included in the criteria of this systematic review. Please note that the new references were updated.
- Line 283-284, "...an increase in serotonin and norepinephrine concentrations...". The physiological effects of the two hormones are usually opposite, so how would they improve mood? Elaborate.
Authors: Dear Reviewer, we have tried to clarify the text in the manuscript. However, some neuroendocrine adaptations associated with body mind exercise and Tai Chi have revealed positive effects on the symptoms presented for the populations with fibromyalgia, namely in terms of improved physical well-being. The norepinephrine neurotransmitter is known to be involved in a range of physiological and psychological processes, and dysfunctions of this neurotransmitter system have been implicated in a range of psychological and psychiatric disorders (Brunello et al, 2003). The norepinephrine modulates drive and energy and exerts a fine regulation of specific processes including learning, memory, sleep, arousal and adaptation (Montgomery, 1997). The norepinephrine and serotonin are involved in the modulation of arousal and mood and have been related to a variety of affective functions as well as associated clinical dysfunctions (Singleton et al, 2014; Nutt 2002).
Singleton, O., Hölzel, B. K., Vangel, M., Brach, N., Carmody, J., & Lazar, S. W. (2014). Change in brainstem gray matter concentration following a mindfulness-based intervention is correlated with improvement in psychological well-being. Frontiers in human neuroscience, 8, 33.
Nutt, David J.. The neuropharmacology of serotonin and noradrenaline in depression. International Clinical Psychopharmacology 17():p S1-S12, June 2002.
https://doi.org/10.1097/00004850-200206001-00002
Brunello, N., Blier, P., Judd, L. L., Mendlewicz, J., Nelson, C. J., Souery, D., ... & Racagni, G. (2003). Noradrenaline in mood and anxiety disorders: basic and clinical studies. International clinical psychopharmacology, 18(4), 191-202.
Montgomery SA (1997). Reboxetine: additional benefits to the depressed patient.
J Psychopharmacol 11:S9–S15
- The logic in the discussion section is rather hard to follow and it is difficult to understand how the conclusions were reasoned out. Need to adda clear synthesis after discussing each point.
Authors: Dear Reviewer, as per your recommendation, the structure of the discussion section was organized so that a clear summary of the discussion was presented following the analysis of each topic.
Reviewer 2 Report
The authors conscientiously developed, selected and summarized the results of many scientific reports on fibromyalgia disease (FM).
1. Some of the exclusion criteria regarding scientific preference for the criterion of random assignment to study groups may be objected to. Well, random selection is not advisable in the case of, for example, evaluating the effectiveness of strength training, because the effects may be disturbed by individual differences in anthropometric conditions, characteristics of the muscular system or even the training experience of the subjects.
2. Concluding on the effectiveness of the use of moderate training loads to improve the health condition disturbed by fibromyalgia is also unjustified. No attempt was made to unify these loads, expressed differently for strength, endurance or flexibility training. Therefore, in scientific work, colloquial terms such as "light or moderate intensity" cannot be used, in addition without distinguishing and defining the basic components of the load - volume and intensity.
Author Response
Reviewer 2
Comments and Suggestions for Authors
The authors conscientiously developed, selected and summarized the results of many scientific reports on fibromyalgia disease (FM).
- Some of the exclusion criteria regarding scientific preference for the criterion of random assignment to study groups may be objected to. Well, random selection is not advisable in the case of, for example, evaluating the effectiveness of strength training, because the effects may be disturbed by individual differences in anthropometric conditions, characteristics of the muscular system or even the training experience of the subjects.
Authors: Dear reviewer, we agree with your concern. Nonetheless, in addition to presenting sample inclusion criteria, the included RCT studies showed in their own studies that no differences between any groups existed. Thus, it seems correct the keep the criteria.
- Concluding on the effectiveness of the use of moderate training loads to improve the health condition disturbed by fibromyalgia is also unjustified. No attempt was made to unify these loads, expressed differently for strength, endurance or flexibility training. Therefore, in scientific work, colloquial terms such as "light or moderate intensity" cannot be used, in addition without distinguishing and defining the basic components of the load - volume and intensity.
Authors: Thank you for raising this point. We understand your point, but we sustained our option based on the ACSM guidelines for exercise prescription which also used this type of expressions. Nonetheless, we modify some parts of discussion and conclusion to clarify this topic, accordingly.
Reviewer 3 Report
This study provides a systematic overview of the effects of complex training programs in fibromyalgia patients and presents an effective exercise program for improving symptoms and quality of life in fibromyalgia patients. On the other hand, the present study requires revision and reconsideration of the following points.
(1) Finally, 13 papers were included in this analysis. Is a sample size of 13 papers a reasonable number for a systematic review? The validity of the sample size should be described.
(2) Analysis and discussion of race should be included.
(3) Differences in the effectiveness of exercise programs based on the number of years of diagnosis should be discussed.
(4) The mechanism by which the combined exercise program provides improvement in pain should be discussed.
Author Response
Comments and Suggestions for Authors
This study provides a systematic overview of the effects of complex training programs in fibromyalgia patients and presents an effective exercise program for improving symptoms and quality of life in fibromyalgia patients. On the other hand, the present study requires revision and reconsideration of the following points.
Authors: Thank you very much for your comments and suggestions.
(1) Finally, 13 papers were included in this analysis. Is a sample size of 13 papers a reasonable number for a systematic review? The validity of the sample size should be described.
Authors: We appreciate your question. To the best of authors knowledge, there is not consensus on a minimum number to conduct a systematic review. Even so, the topic of our work is quite embracing and our goal was to provide specific information for fitness professionals and exercise physiologist about the training prescription. For instance, ACSM just provide general descriptions while this study presents several details. Finally, we would like to mention that there are other systematic reviews with lower number of articles included. Here are some examples with just 5 articles and eight, respectively.
Martins, A.D., Oliveira, R., Brito, J.P., Costa, T., Silva, J., Ramalho, F., Santos-Rocha, R., Pimenta, N. (2022). Effect of exercise on phase angle in cancer patients: a systematic review. The Journal of Sports Medicine and Physical Fitness. 62(9), 1255–1265. https://doi.org/10.23736/S0022-4707.21.12727-6
Oliveira, R.; Brito, J.P.; Moreno-Villanueva, A.; Nalha, M.; Rico-González, M.; Clemente, F.M. Reference Values for External and Internal Training Intensity Monitoring in Young Male Soccer Players: A Systematic Review. Healthcare 2021, 9, 1567. https:// doi.org/10.3390/healthcare9111567
(2) Analysis and discussion of race should be included.
Authors: Dear reviewer, thank you for raising this point. Indeed, this may have some influence in the results and the authors would like to add this information. However, when analysing the 13 articles, only one study included this information (Wang et al., 2018) [15] while the others did not present any information. For that reason, we did not add it in the results, but we suggest to consider it as a recommendation for future research.
(3) Differences in the effectiveness of exercise programs based on the number of years of diagnosis should be discussed.
Authors: Thank you for the suggestion. We revised the discussion and added this topic. Still, it is relevant to note that some articles did not provide that information and only stated that fibromyalgia was clinically diagnosed.
(4) The mechanism by which the combined exercise program provides improvement in pain should be discussed.
Authors: Dear reviewer, thank for the suggestion. Discussion was revised and now it provided the requested information. Thank you
Reviewer 4 Report
First of all, congratulations for the work done, then I will mention a number of changes and recommendations in order to obtain clearer and more accurate information.
-Comments on the abstract:
· The correct verb is “was” not “were” on line 18.
· Do not use abbreviations on the abstract (FM in line 20) moreover if you have not defined it before.
- Comments on the introduction:
· The abbreviation of fibromyalgia must be made the first time it appears in the text (line 35 instead of line 41).
· Once defined the abbreviation of fibromyalgia, you must always use the abbreviation FM (line 44), you must revise all the manuscript.
· Missing dot on line 50 before "However", and on line 55 after references.
- Comments on material and methods
· Why do you delete articles that are not in English?
· Again, you make a mistake with the abbreviations, PEDro is defined before and you write it again without abbreviations.
- Comments on results:
- Table 1.
- Abbreviation must be defined in the order of appearance in the table.
- M is not the abbreviation of mean, XÌ… must be used.
- Line 241, you are repeating the number of the table, this one should be Table 3.
- Table 1 that must be Table 3, the foot is incorrect, there are some abbreviations that not appear in the table.
-General comments:
In figure 1, you talked about 13 articles included in the quantitative analysis, there are not any quantitative analysis as you are not doing any meta-analysis.
Another different matter is that you attempt to make a quantitative assessment of the duration of the interventions. It would be very interesting, and I'm sure you can do it, to rewrite this paper including the meta-analysis section.
Author Response
Reviewer 4
Comments and Suggestions for Authors
First of all, congratulations for the work done, then I will mention a number of changes and recommendations in order to obtain clearer and more accurate information.
-Comments on the abstract:
- The correct verb is “was” not “were” on line 18.
- Do not use abbreviations on the abstract (FM in line 20) moreover if you have not defined it before.
Authors: : Dear reviewer, both comments were addressed accordingly.
- Comments on the introduction:
- The abbreviation of fibromyalgia must be made the first time it appears in the text (line 35 instead of line 41).
- Once defined the abbreviation of fibromyalgia, you must always use the abbreviation FM (line 44), you must revise all the manuscript.
- Missing dot on line 50 before "However", and on line 55 after references.
Authors: Dear reviewer, all comments were addressed accordingly. Thank you very much for your attention.
- Comments on material and methods
- Why do you delete articles that are not in English?
Authors: Dear reviewer, thank you for your observation. Indeed, we only selected articles written in English because is the universal language of science, according a previous study there is no evidence of a systematic bias of the use of the language restriction ( Morrison, et al 2012). Thank you
Morrison A, Polisena J, Husereau D, Moulton K, Clark M, Fiander M, Mierzwinski-Urban M, Clifford T, Hutton B, Rabb D. The effect of English-language restriction on systematic review-based meta-analyses: a systematic review of empirical studies. Int J Technol Assess Health Care. 2012 Apr;28(2):138-44. doi: 10.1017/S0266462312000086. PMID: 22559755.
- Again, you make a mistake with the abbreviations, PEDro is defined before and you write it again without abbreviations.
Authors: Dear reviewer, PEDro was only mentioned in abstract. After that, it was only reported in section 2.1. Abbreviation was removed from the abstract accordingly with your previous comments. Thank you
- Comments on results:
- Table 1.
- Abbreviation must be defined in the order of appearance in the table.
- M is not the abbreviation of mean, XÌ… must be used.
Authors: both comments were amended. Thank you
- Line 241, you are repeating the number of the table, this one should be Table 3.
- Table 1 that must be Table 3, the foot is incorrect, there are some abbreviations that not appear in the table.
Authors: Dear reviewer, all abbreviations and the numbers were amended. Thank you
-General comments:
In figure 1, you talked about 13 articles included in the quantitative analysis, there are not any quantitative analysis as you are not doing any meta-analysis.
Another different matter is that you attempt to make a quantitative assessment of the duration of the interventions. It would be very interesting, and I'm sure you can do it, to rewrite this paper including the meta-analysis section.
Authors: We understand your comment, however we decided not to add meta-analysis to this study. However, there were no identical training programs, and a meta-analysis would just create a bias when interpreting all results. Even so, we provided a limitation and an additional suggestion for future as follows: “Finally, when conducting a future systematic review study on this population, a meta-analysis would be interesting to strengthen the results of the same intervention type which was not possible in the present study.” This information of was added in the last part of discussion. Thank you
Round 2
Reviewer 1 Report
No futher comments.
Author Response
Reviewer 1
No futher comments.
Authors: Dear Reviewer, we would like to thank you for the time and effort devoted to reviewing our manuscript, which greatly contributed to improving its overall quality.
Reviewer 4 Report
I am pleased to see the changes you have made; they have greatly improved the quality of the work. However, I believe there are still a few minor details to be corrected:
Table 1. Characteristics of The Exercise Programs Present in the Systematic Review. Should be named Table 4.
The tables "Characteristics of the articles included in the systematic review" and "Characteristics of The Exercise Programs Present in the Systematic Review" should follow the same format with abbreviations. Either use abbreviations in both tables or in neither, in order to standardize the format of the work.
Other than that, as I mentioned before, I am pleased with the changes made. By making these small corrections, I believe the work could be accepted.
